# Environmental Optimization of Precast Concrete Beams Using Fibre Reinforced Polymers

**R. R. L. (Rick) van Loon, Ester Pujadas-Gispert \*, S. P. G. (Faas) Moonen and Rijk Blok**

Eindhoven University of Technology, Eindhoven, The Netherlands; r.r.l.v.loon@alumnus.tue.nl (R.R.L.(R.)v.L.); S.P.G.Moonen@tue.nl (S.P.G.(F.)M.); R.Blok@tue.nl (R.B.)

\* Correspondence: e.pujadas.gispert@tue.nl; Tel.: +31-40-247-6047

**Abstract:** Increasing importance is being attached to materials in the life-cycle of a building. In the Netherlands, material life-cycle assessments (LCA) are now mandatory for almost all new buildings, on which basis the building is then awarded a building environmental performance or MPG [Milieuprestatie Gebouwen] score. The objective of this study is to reduce the environmental–economic (shadow) costs of precast reinforced concrete (RC) beams in a conventional Dutch office building, thereby improving its MPG score. Two main optimizations are introduced: first, the amount of concrete is reduced, designing a cavity in the cross-section of the beam; second, part of the reinforcement is replaced with a fibre reinforced polymer (FRP) tube. The structural calculations draw from a combination of several codes and FRP recommendations. Hollow FRP-RC beams (with an elongated oval cavity), and flax, glass, and kenaf fibre tubes yielded the lowest shadow costs. In particular, the flax tube obtained shadow costs that were 39% lower than those of the hollow RC beam (with an elongated oval cavity); which also contributed to decreasing the shadow costs of other building components (e.g., facade), thereby reducing the MPG score of the building. However, this study also shows that it is important to select the right type of FRP as hemp fibre tubes resulted in a 98% increase in shadow costs.

**Keywords:** prefabricated precast concrete beams; cross-sectional design; fibre reinforced polymer; MPG score; shadow cost; carbon; glass; flax fibres; kenaf; FRP

## 1. Introduction

The building sector is now recognized as having a higher potential than any other to deliver quick, in-depth and cost-effective greenhouse gas (GHG) mitigation. The environmental footprint of the building sector by percentage consumption comprises 40% energy, 30% raw materials, 25% solid waste, 25% water resources, and 12% land [1]. This sector is also responsible for the generation of 860 million tons of waste, which is 34% of all waste produced in the European Union [2]. If climate change is to be slowed down, it is therefore crucial to reduce the environmental burden of the building sector.

The energy used during the construction and demolition of a building, also called the embodied phase, is gaining importance. This phase is normally expected to represent 10%–20% of the life-cycle energy of the building, while the remaining 80%–90% is expected to be consumed during the operational phase of the building (heating, cooling, etc.) [1]. Nevertheless, some studies have shown that the embodied phase in very low energy buildings can account for over 50% of the life-cycle energy of a building [3,4]. Buildings are beginning to adhere to higher energy efficiency standards, and as a result, the energy consumption during the operational phase is significantly reduced. In addition, their facades and technical installations require more materials, some of which are highly-energy intensive [5]. It is therefore increasingly important to assess the embodied phase of the building, and especially to consider its construction, when the aim is to improve the life cycle of the building.

In the construction of a building, materials normally represent the highest environmental burdens; in particular, concrete because of the large amounts that are used [6]. Concrete is commonly used in construction because its raw materials—cement, aggregates and water—are cheap and widely available [7]. Cement, which normally accounts for only 12% of concrete mass, is responsible for most of its greenhouse gases [8]. The 1.6 billion tons of the world's annual production of cement accounts for around 7% of the global loading of the atmosphere with carbon dioxide. Other concrete components comprise large quantities of raw materials and fuel, often resulting in extensive deforestation and top-soil loss [8].

There are several innovations designed to reduce the environmental impacts of reinforced concrete (RC) structures that are focused on improving cement mixtures and concrete mechanical properties. However, other approaches are also possible. For instance, designing a cavity in the cross-section of a beam can significantly reduce the amount of concrete required to construct it, although certain structural considerations must then be taken into account [9]. Furthermore, a concrete structure with replaceable components might facilitate building repair and reuse of its components, thereby extending its life span and minimizing its environmental impact [10]. Similarly, (partial or total) replacement of conventional steel bar reinforcements by other types (e.g., fibres, strips, sheets, etc.) of reinforcement and materials (e.g., basalt, polypropylene, etc.) might improve the sustainability of some RC structures [11–13], and their mechanical performance. One example is the fibre reinforced polymer tube reinforced concrete (FRP-RC) beam. It basically consists of an RC beam, with or without a cross-sectional cavity, and an FRP tube substituting part of the conventional steel bar reinforcement. The results show that these beams can have a significantly higher overall strength-to-weight ratio in comparison with conventional RC beams [14].

There are many types of FRP reinforcement [15] that are normally composed of a polymeric matrix and fibres. The polymer material is usually an epoxy or vinyl ester, but bio-based resins are also gaining ground. Fibers can moreover be classified under either synthetic fibres such as carbon and glass, or natural fibres such as hemp and jute [16]. Natural fibres are a renewable resource and their production that involves $CO_2$ absorption while returning oxygen to the atmosphere requires little energy. Among others, one drawback of natural fibres is their durability, although it can be improved considerably with treatment, and the great variability of properties that can change, depending on several factors such as harvesting [17]. FRP composites can serve as reinforcements for new structural members (internal reinforcements) and for strengthening existing members (externally bonded reinforcements) [16].

Increasing importance is attached to materials in the life-cycle of works. Therefore, regulations are starting to take their impact into account. Since January 2013, in The Netherlands [18], it is compulsory to submit a calculation of the environmental impact embodied in building materials when applying for a building permit with a floor area larger than 100 m$^2$ [19]. As a result, the building receives an environmental performance or MPG [Milieuprestatie Gebouwen] score [19]. This calculation must follow the Dutch assessment method [20,21], which provides guidelines for calculating the material-based environmental performance of construction and civil engineering works over their service life. This assessment is closely connected to the National Environmental Database (NMD) [22]. The environmental impacts are reported under eleven categories and are transformed into economic costs by means of weighting factors, commonly referred to as shadow costs (Table 1).

**Table 1.** Shadow cost for each environmental impact category [20].

| Environmental Impact Categories | Unit | Shadow Cost (€/kg eq) |
|---|---|---|
| Depletion of abiotic resources (excluding fossil fuels) (ADP) | Sb eq | 0.16 |
| Depletion fossil fuels (ADP) | Sb eq[6] | 0.16 |
| Global warming for 100 years (GWP 100 j.) | $CO_2$ eq | 0.05 |
| Depletion ozone layer (ODP) | CFK-11 eq | 30 |
| Photochemical oxidant creation (POCP) | $C_2H_4$ eq | 2 |
| Acidification (AP) | $SO_2$ eq | 4 |
| Eutrophication (EP) | $PO_4$ eq | 9 |
| Human toxicity (HTP) | 1,4-DCB eq | 0.09 |
| Fresh water aquatic eco toxicity (FAETP) | 1,4-DCB eq | 0.03 |
| Marine aquatic eco toxicity (MAETP) | 1,4-DCB eq | 0.0001 |
| Terrestrial eco toxicity (TETP) | 1,4-DCB eq | 0.06 |

The shadow cost of a material is calculated by adding the shadow costs of its environmental impact categories. The MPG score of a building is calculated by adding all shadow costs of its materials, divided by the gross floor area and its life span (1).

$$\text{MPG} = \frac{\sum(\text{Shadow cost each material})}{\text{Gross floor area x Building life span}} \leq \frac{1.00}{\text{m}^2} \text{ Gross floor area x year} \qquad (1)$$

Nowadays, the average MPG score of a Dutch office building is around €0.48/gross m$^2$ year, which is far below the required €1.00/gross m$^2$ year [19]. Despite the fact that almost every building in the Netherlands satisfies this requirement, the MPG score might be increased when the new EPC [Energieprestatie] [23], which governs the energy building consumption, enters into force in 2020. As previously explained, the environmental impact embodied in the materials of energy efficient buildings tends to be higher. Besides, each building component (e.g., foundation, floors, structure, facade, etc.) has a different weight in the MPG score (Figure 1). Although structure only accounts for 2% of the MPG score, its optimization might lead to reductions of the shadow costs of other components, such as the facade and installations that have a greater weight, and which therefore obtain a better MPG score.

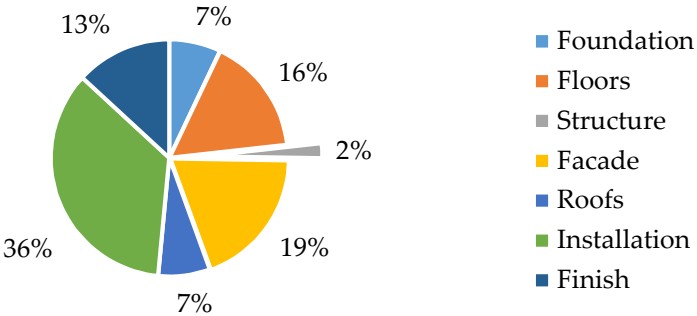

**Figure 1.** Contribution of building components in an MPG (Milieuprestatie Gebouwen) score of a building [19].

However, the literature on the optimization of RC structures from an environmental viewpoint is scarce and is very rarely based on an MPG score. Hence, the focus of this article is on the optimization and the improvement of the environmental impact of precast concrete beams in a conventional Dutch office building to improve its MPG score. For this purpose, two optimizations were performed: first, the amount of concrete was reduced, by designing a cavity in the cross-section of the beam; second, the conventional reinforcement was partly replaced by a fibre reinforced polymer (FRP) tube. The specific objectives were as follows: (1) to calculate a conventional precast concrete beam reinforced

with steel bars; (2) to optimize it, by designing a cavity in the cross-section; (3) to calculate the amounts of FRP required with several polymer matrixes and fibres; (4) to calculate the shadow cost of the alternatives and to analyze them, so as to define specific design conclusions and recommendations.

## 2. Materials and Methods

In this section, the case study is presented (Section 2.1) together with the data required to conduct the structural analysis of study beams (Section 2.2), the optimization of both the concrete (Section 2.3) and the reinforcement of the beams (Section 2.4), and the calculation of their shadow costs (Section 2.5).

### 2.1. Case Study

The precast concrete beam is part of a conventional Dutch office building. Figure 2 shows the structure of the building, which consists of a precast concrete frame (beams and pillars) and hollow core slabs. The beams have a span of 7.2 m and are connected to the concrete pillars. The hollow core slabs have a thickness of 320 mm, a span of 10 m, and they are placed over the beams. In addition, common loads for Dutch offices are applied [24–27]. The distributed load over the beams is 105.5 kN/m. The structural materials are concrete (C60/75) and steel (B500B).

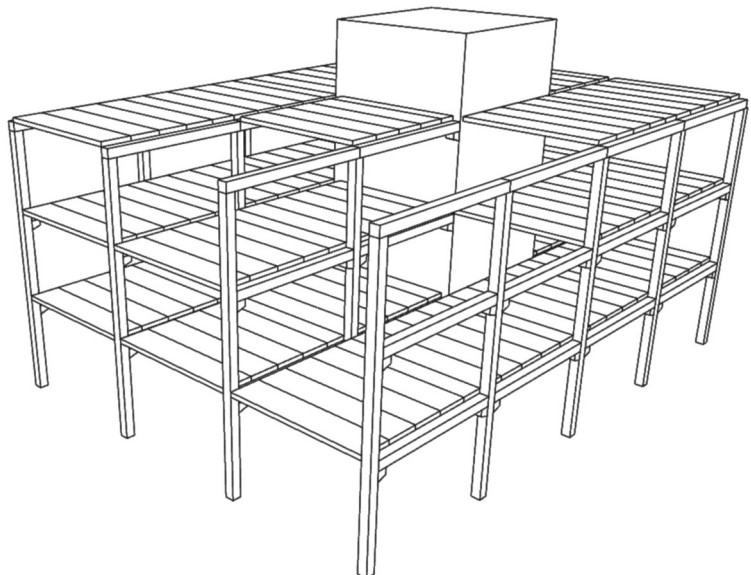

**Figure 2.** Conventional Dutch office building structure.

### 2.2. Structural Analysis

The structural design was based on a combination of several codes and recommendations. All the beams under study were designed in accordance with the Eurocode 2 [28,29], and their steel bar reinforcements were calculated with SCIA Engineer [30]. The FRP tube of FRP-RC beams was calculated with FIB Bulletin 14 [31], CB4 Chapter 9—Strengthening of existing concrete structures [32] and the Dutch CUR Recommendation 91 [33]. These codes were rewritten with Mathematica code, in order to adapt them to the case study. A brief description of each chapter in the Mathematica file is shown in the Table A1 of the Appendix A, and further explanations can be found in [34]. The following considerations were made in order to calculate the FRP-RC beams:

- First, FRP recommendations focus on improving existing RC structures by applying carbon fibre strips or sheets. Nevertheless, a newly built beam was used in the case study, and therefore, there was not a loading history. Consequently, the determination of the initial strain and deflection were not necessary.

- Second, the difficulty of applying the FRP sheets can create a gap between the end support and the start of the FRP. Shear forces are determinant in the gap, hence the crack behaviour should be

checked. Nevertheless, as the study beam under consideration was wrapped all length in a fibre tube [14], no checks were required. The FRP tube was finished with a rough surface. This surface would ensure a fully bond between the FRP and concrete, greatly minimizing any possibility of early rupture [35].

*2.3. Concrete Optimization*

The concrete optimization process consisted of designing a cavity in the cross-section. The overall structural behaviour was checked, as well as cracking in the compression zone, which is a determining failure mechanism in hollow cross-sections [9]. Both circular and rectangular shapes were considered for the cavity. In addition, an elongated oval cavity was designed, in order to minimize the concrete volume of the cross-section. The larger oval cavity in no case entered the compression zone under the maximum bending moment (Figure 3b).

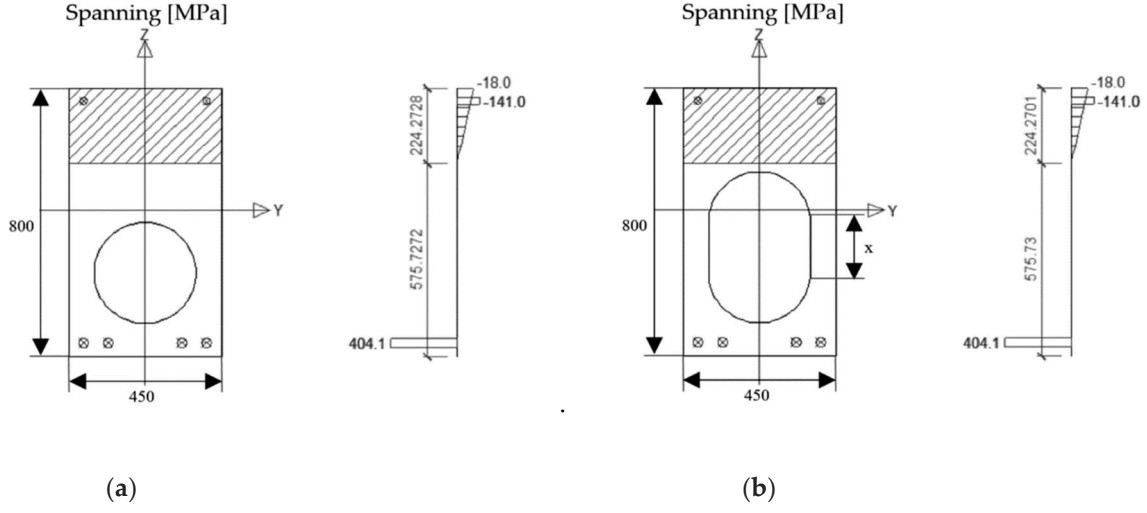

(**a**)　　　　　　　　　　　　　　　　　　　　　　　　　(**b**)

**Figure 3.** Stress development in a beam cross-section with a circular cavity (**a**) and in a beam cross-section with an elongated oval cavity (**b**), both calculated in SCIA Engineer [30]. Terminology: straight part of the cavity (x).

*2.4. Reinforcement Optimization*

Then, the reinforcement of the beam with the most highly optimized amounts of concrete was improved. To do so, part of the steel bar reinforcement was substituted by an FRP tube. Several tubes were tested, made of different fibres and matrixes (Table 2) [17]. On this point, it is worth mentioning that the selection of the matrix conditions the mechanical properties and the fibre content of the FRP, which in turn conditions the amount of resin that is required.

After selecting the fibre composites of the study, their properties were introduced into the Mathematica file, in order to calculate the thickness of the FRP tubes that would satisfy the Eurocode 2 [28,29] specifications. Then, the volumes of FRP for each beam were calculated. Finally, those volumes were multiplied by each material density, in order to obtain the kilograms of FRP for each beam.

**Table 2.** Mechanical properties of selected synthetic and natural fibre composites.

| Fibers | Density (kg/m$^3$) | Matrix (-) | Fiber Content (%) | Tensile Strength (MPa) | Young's Modulus (GPa) | Failure Strain (%) |
|---|---|---|---|---|---|---|
| Carbon | 1800 | Epoxy | ±60 | 1043 | 60.66 | 1.72 |
| E-glass | 2550 | Epoxy | 59 | 483 | 33 | 1.46 |
| Flax | 1400 | PP (aligned) | 72 | 321 | 29 | 1.07 |
| Jute | 1460 | PP | 60 | 74 | 11 | 0.67 |
| Hemp | 1393 | PLA (biaxial) | 65 | 165 | 17 | 0.97 |
| Kenaf | 1200 | PLA (aligned) | 80 | 223 | 23 | 0.97 |

*2.5. Calculation of Shadow Costs*

Once the kilograms of materials (concrete, steel and FRP) had been determined, these amounts were multiplied by the shadow costs of each material to obtain the total shadow costs of hollow beams (RC and FRP-RC). Nevertheless, shadow costs for the FRP in this study were not found. Hence, shadow costs for fibres and resins were separately retrieved from the Vogtländer [36] and the TUDelft [37] databases. In addition, several assumptions were made, in order to calculate the shadow costs of each beam. First, the compressive strength of the concretes used in this research (C60/75 and C70/85) were higher than those found in the shadow cost databases (for instance C20/35). Second, the transportation and production costs of the beams were not included in the study. While these assumptions might change the shadow costs of the beams, their effects on the findings of this research would be minimal.

## 3. Results and Discussion

The results are organized with the following sub-sections: optimization of concrete beam cross-section (Section 3.1), optimization of beam reinforcement (Section 3.2), and calculation of shadow costs of beams (Section 3.3).

*3.1. Optimization of Concrete Beam Cross-Section*

The conventional RC beam (with no cavity in the cross-section) resulted in an $800 \times 450 \text{ mm}^2$ concrete cross-section, with 4Ø28 steel reinforcing bars, and Ø12 stirrups. Two specimens were considered for the optimization of the cross-section, with a similar cavity area, but with either a rectangular or a circular shape. The cracking moment of the beam was limited at 4%. In addition, the yield moment ($M_y$), of the beam with a circular cavity was 15% higher than the $M_y$ of the beam with a rectangular cavity. A difference that is explained by the reinforcement of the circular cavity that had more concrete around it than the rectangular cavity. The distribution of shear forces through the cross-section was therefore more homogeneous around the circular cavity than it was around the rectangular cavity. Finally, there was also a small yet perceptible difference in the ultimate moment ($M_u$) at failure. The beams with a rectangular cavity had an ultimate load that was around 7% smaller than the beam with the circular cavity. This small difference could be explained by the yield moment ($M_y$) after yielding the flexural strength of the beam, the beam's resistance depends on the concrete. In addition, a second optimization of the cross-section was performed. As can be seen in Figure 3a, there is still space to extend the cavity up to the compression zone (shaded area). Figure 3b shows the beam cross-section with an elongated oval cavity. By applying this extension of 150 mm to the Ø300 mm cavity, the concrete cavity area was enlarged by 60%.

*3.2. Optimization of Beam Reinforcement*

The reinforcement of the enhanced hollow RC beam (Figure 3b) was optimized. Part of the conventional steel bar reinforcement was replaced by an FRP-RC tube of different composites. Table 3 presents the main characteristics and mechanical properties of the hollow beams (RC and FRP-RC). As can be seen, the concrete compressive strength of the natural FRP-RC beams was increased (from C60/75 to C70/85), because the natural FRP fibres have a low ultimate strain. The cross-sectional areas of concrete, steel and FRP of the beams are shown at the end of the table.

**Table 3.** Main characteristics and mechanical properties of hollow beams.

| Hollow Beams | (Steel) RC | Carbon FRP-RC | E-glass FRP-RC | Flax FRP-RC | Jute FRP-RC | Hemp FRP-RC | Kenaf FRP-RC |
|---|---|---|---|---|---|---|---|
| Fibre | - | Carbon | E-glass | Flax | Jute | Hemp | Kenaf |
| Resin | - | Epoxy | Epoxy | PP | PP | Epoxy | PLA |
| Tensile strength | - | 1043 MPa | 483 MPa | 321 MPa | 74 MPa | 105 MPa | 223 MPa |
| Young's modulus | - | 60.66 GPA | 33 GPA | 29 GPA | 11 GPA | 90 GPa | 23 GPA |

**Table 3.** *Cont.*

| Hollow Beams | (Steel) RC | Carbon FRP-RC | E-glass FRP-RC | Flax FRP-RC | Jute FRP-RC | Hemp FRP-RC | Kenaf FRP-RC |
|---|---|---|---|---|---|---|---|
| Fiber content | - | 60% | 59% | 72% | 60% | 50% | 80% |
| FRP bending | - | 1.36 | 3.75 | 6.30 | 21.63 | 19.26 | 10.42 |
| FRP shear | - | 0.40 | 1.11 | 2.34 | 9.98 | 6.63 | 3.82 |
| Concrete | C60/75 | C60/75 | C60/75 | C70/85 | C70/85 | C70/85 | C70/85 |
| Tension bars | 2Ø25 | 2Ø25 | 2Ø25 | 2Ø25 | 2Ø25 | 2Ø25 | 2Ø25 |
| Comp. bars | 4Ø28 | 3Ø16 | 3Ø16 | 3Ø16 | 4Ø16 | 3Ø16 | 4Ø16 |
| Height | 800 mm | 800 mm | 700 mm | 700 mm | 800 mm | 700 mm | 650 mm |
| Width | 450 mm | 450 mm | 450 mm | 400 mm | 400 mm | 400 mm | 400 mm |
| Straight part of the cavity [1] | 150 mm | 150 mm | 125 mm | 100 mm | 0 mm | 100 mm | 0 mm |
| **Material Areas in mm$^2$** | | | | | | | |
| Concrete area | 244,000 | 242,729 | 205,229 | 177,729 | 247,528 | 177,729 | 172,528 |
| Steel area | 2916 | 1585 | 1585 | 1585 | 1786 | 1585 | 1786 |
| FRP area | 0.00 | 1855.49 | 4958.71 | 8371.64 | 34,134.80 | 25,200.60 | 13,465.30 |

[1] *x* in Figure 3.

There are multiple advantages to the use of FRP as reinforcement: first of all, FRP materials can be chosen according to their environmental impact. Second, in hollow beams, the space to place the reinforcement is sometimes reduced; and therefore, minimum concrete cover widths might not be guaranteed. Concrete covers protect the reinforcement from corrosion and help to ensure the durability of the beam. In this respect, the use of FRP tubes reduces the quantity of steel reinforcement and ensures the concrete widths. Besides, with an FRP tube, no framework is required to build the beam, which is beneficial for the life cycle of the beam.

*3.3. Shadow Costs of Beams*

Table 4 shows the total shadow costs of hollow beams (indicated in bold) and the data for their calculation. As can be seen, material amounts vary depending on the beam, which is explained by the different properties of the FRP that not only contribute to the amount of FRP, but also to the quantities of steel and concrete.

**Table 4.** Shadow cost calculation of hollow beams.

| | Hollow Beams | Material | Density (kg/m$^3$) | FRP Content (%) | Total Volume (m$^3$) | Total Mass (kg) | Shadow Costs (€/kg) | Total Shadow Costs (€/beam) |
|---|---|---|---|---|---|---|---|---|
| 1 | RC | Concrete | 2400.00 | | 2592.00 | 6220.80 | 0.02 | 124.60 |
| | | Reinforcement steel | 7800.00 | | 0.02 | 160.91 | 0.17 | 27.37 |
| | | | | | 2.61 | 6381.71 | | **151.98** |
| 2 | Carbon FRP-RC | Concrete | 2400.00 | | 1.75 | 4194.36 | 0.02 | 84.01 |
| | | Reinforcement steel | 7800.00 | | 0.01 | 89.01 | 0.17 | 15.14 |
| | | Carbon + epoxy | 1800.00 | 100.00 | 0.01 | 24.05 | 2.64 | 63.48 |
| | | | | | 1.77 | 4307.42 | | **162.64** |
| 3 | E-glass FRP-RC | Concrete | 2400.00 | | 1.48 | 3546.36 | 0.02 | 71.03 |
| | | Reinforcement steel | 7800.00 | | 0.01 | 89.01 | 0.17 | 15.14 |
| | | E-glass | 2550.00 | 59.00 | 0.04 | 53.72 | 0.10 | 5.37 |
| | | Epoxy resin | 1150.00 | 41.00 | 0.04 | 16.83 | 1.03 | 17.28 |
| | | | | | 1.56 | 3705.92 | | **108.82** |
| 4 | Flax FRP-RC | Concrete | 2400.00 | | 1.28 | 3071.16 | 0.02 | 61.52 |
| | | Reinforcement steel | 7800.00 | | 0.01 | 89.01 | 0.17 | 15.14 |
| | | Flax | 1400.00 | 58.00 | 0.06 | 48.94 | 0.23 | 11.23 |
| | | PP (R) resin | 855.00 | 42.00 | 0.06 | 21.64 | 0.24 | 5.19 |
| | | | | | 1.41 | 3230.76 | | **93.08** |

**Table 4.** *Cont.*

| | Hollow Beams | Material | Density (kg/m³) | FRP Content (%) | Total Volume (m³) | Total Mass (kg) | Shadow Costs (€/kg) | Total Shadow Costs (€/beam) |
|---|---|---|---|---|---|---|---|---|
| 5 | Jute FRP-RC | Concrete | 2400.00 | | 1.78 | 4277.28 | 0.02 | 85.67 |
| | | Reinforcement steel | 7800.00 | | 0.01 | 100.30 | 0.17 | 17.06 |
| | | Jute | 1460.00 | 60.00 | 0.26 | 215.30 | 0.22 | 47.36 |
| | | PP (R) resin | 855.00 | 42.00 | 0.26 | 88.26 | 0.24 | 21.18 |
| | | | | | 2.29 | 4681.14 | | **171.28** |
| 6 | Hemp FRP-RC | Concrete | 2400.00 | | 1.28 | 3071.16 | 0.02 | 61.52 |
| | | Reinforcement steel | 7800.00 | | 0.01 | 89.01 | 0.17 | 15.14 |
| | | Hemp | 1393.00 | 50.00 | 0.18 | 126.38 | 0.88 | 111.21 |
| | | Epoxy resin | 1210.00 | 50.00 | 0.18 | 109.77 | 1.03 | 112.66 |
| | | | | | 1.65 | 3396.32 | | **300.53** |
| 7 | Kenaf FRP-RC | Concrete | 2400.00 | | 1.24 | 2981.28 | 0.02 | 59.72 |
| | | Reinforcement steel | 7800.00 | | 0.01 | 100.30 | 0.17 | 17.06 |
| | | Kenaf | 1200.00 | 80.00 | 0.10 | 93.07 | 0.17 | 15.45 |
| | | PLA (R) resin | 1210.00 | 20.00 | 0.10 | 23.46 | 0.21 | 5.02 |
| | | | | | 1.45 | 3198.12 | | **97.25** |

Figure 4 shows the shadow costs of the different beams. The RC beam has an average shadow cost of €151.98, while the FRP-RC beam with carbon fibres has a slightly higher cost (€162.64). This difference is explained by the high environmental impact of the carbon FRP, even though it has good mechanical properties. In addition, the FRP concrete beam with hemp fibres has the highest shadow cost (€300), which means 98% more shadow costs than the hollow RC beam. This difference is because hemp has poor mechanical properties, and as a result, the FRP tube was thicker. In this case, the hemp and epoxy resin accounted for 75% of the shadow cost of the hemp FRP-RC beam.

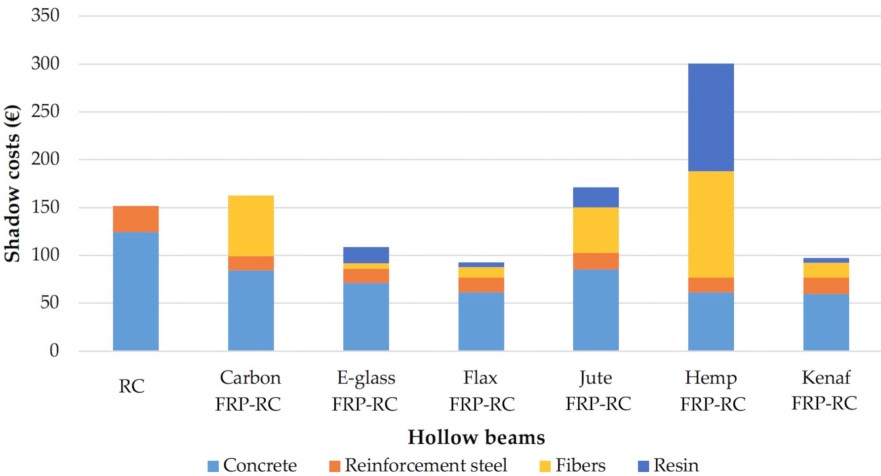

**Figure 4.** Shadow costs of hollow reinforced concrete (RC) beams.

Figure 5 shows the shadow costs of the FRP-RC beams against the RC beam. As can be seen, the total shadow cost for the E-glass, flax and kenaf FRC beams was reduced by 28%, 39%, and 36% respectively, compared to the RC beam. The use of FRP tubes provided a significant reduction of concrete amounts, and consequently shadow costs. These FRP-RC beams coincided with an FRP with good mechanical properties. In this regard, flax fibre is considered exceptional [38]. The flax FRP-RC beam resulted in the lowest shadow costs. In addition, it is one of the most studied fibres and is harvested in the Netherlands [39]. Moreover, the beam height in the standard situation was 800 mm, which could be decreased to 700 mm by using a flax FRP tube. In this way, each floor could be lowered by 100 mm, amounting to a decrease in height of 1 m in a 10-storey building; reducing the total quantity of materials that are needed for each building component, such as the facade [19].

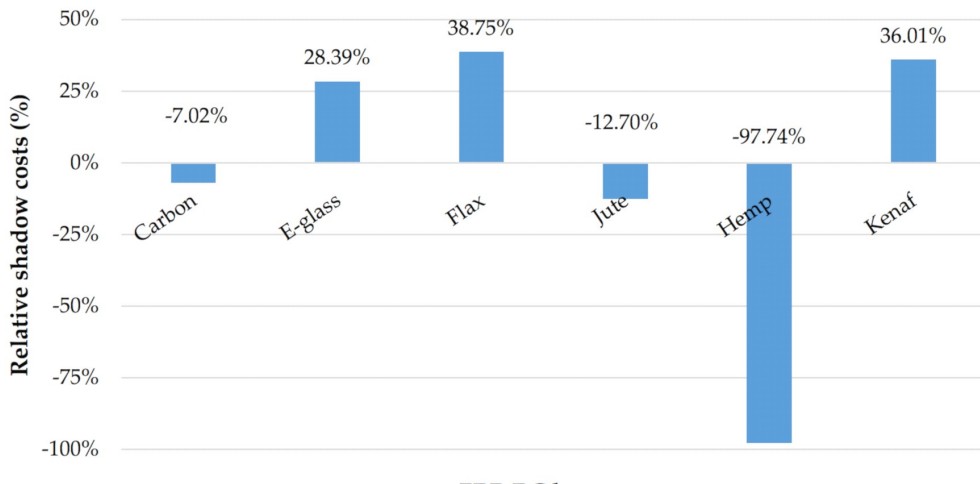

**Figure 5.** Hollow fibre reinforced polymer reinforced concrete (FRP-RC) beams shadow costs compared with hollow reinforced concrete (RC) beam shadow costs.

The use of demountable connections on an FRP concrete beam, means that its replacement or reuse is possible, thereby extending the life span of the building. As a result, the building MPG score would be improved (see Formula (1)). In addition, a demountable structure could reduce the amount of resources as well the waste generation. Nonetheless, the standard metallic connections would require several adjustments for hollow beams. The steel casings are generally connected to the ends of the beam with screw-threads, but this is not possible in these optimized concrete beams because of the cavity in the cross-section. Consequently, the ends of the beam should be filled with concrete during the production process; and in this way, sufficient anchorage length would be ensured, thereby guaranteeing a stable structure.

## 4. Conclusions

The main aim of this research has been to optimize the environmental-economic (shadow) costs of RC beams of a conventional Dutch office building, in order to reduce its MPG score. Hence, two optimizations were introduced: first, the concrete volume was reduced by designing various cavities with different shapes (circular, rectangular and elongated oval) in the cross-section of the beam; and secondly, by optimizing the reinforcement, partly replacing it with a fibre reinforced polymer tube (FRP) of different fibres (carbon, E-glass, flax, jute, hemp, or kenaf). When designing precast RC beams with the intention of reducing the shadow costs, decision-makers should consider several aspects:

- The precast beam with a circular cavity in the cross-section offered a structurally more resistant result than the precast beam with a rectangular cavity of the same area. This strength gain is explained by the fact that the reinforcement of the circular cavity was surrounded with more concrete compared to the rectangular cavity.
- The elongated oval cavity below the compression area was 60% larger compared with the area of the circular cavity. The elongated oval cavity significantly reduced the amount of concrete of the beams.
- The selection of one or another FRP influenced the shadow cost of the beams in several ways. First, each FRP has a specific shadow cost, which depends on the selection of a matrix and fibres. But at the same time, the FRP mechanical properties not only condition the amounts of FRP, but they also condition the amounts of concrete and steel reinforcement.
- FRP tubes composed of E-glass fibres, flax fibres, and kenaf fibres were shown to have the highest potential for decreasing the total shadow costs of the precast RC beams, (28%, 39% and 36% respectively).

- The hollow flax FRP-RC beam had the lowest shadow cost (39% lower) compared to the conventional hollow RC beam because of its good mechanical properties. Its thin tube accounted for a small FRP amount.
- Replacing reinforcement by FRP could result in an adverse environmental impact. For example, the hollow FRP-RC beam with a tube of hemp fibres and epoxy resin accounted for 98% more shadow costs compared to the hollow RC beam. These higher costs were mainly due to the poor mechanical properties of hemp. As a result, the FRP tube needed to be thicker and also required more hemp and epoxy resin. In this example, the hemp and the epoxy resin accounted for 75% of its shadow costs.
- Additionally, the height of the hollow flax FRP-RC beam could be reduced by 100 cm. Consequently, the height of a 10-storey building could be reduced up to 1 m. Thus, not only will the beam optimization reduce the environmental impact of the structure, but it will also minimize other building components such as the facade, thereby enhancing the MPG score of the building.
- A dismountable structure would facilitate the reuse and the replacement of various building components, thereby enlarging the life cycle of the building and improving the MPG score. In addition, waste generation would also be reduced. Special connections for hollow concrete beams would however need to be designed.

Further investigations could verify these findings experimentally, using other combinations of composites. It would also be of interest to optimize concrete beams in the longitudinal direction by the applying cavities. All of these optimizations have the potential to enhance the MPG score of a building.

**Author Contributions:** R.R.L.(R.)v.L. and E.P.-G. were responsible for writing and for revision, and S.P.G.(F.)M. and R.B. acted as the supervisors.

**Funding:** This research received no external funding.

**Acknowledgments:** We gratefully acknowledge Hèrm Hofmeyer for his valuable advice.

**Conflicts of Interest:** The authors declare no conflict of interest.

## Appendix A

**Table A1.** A brief description of structural calculations in the Mathematica file [34].

| Chapter | Description |
|---|---|
| General information | Characteristics of the concrete beam cross-section are introduced: size, cavity area and the amount of compression, and tensile reinforcement. |
| Safety factors | Introduction of material safety factors and load combinations [24,25]. |
| Material grades | Introduction of material grades of concrete, steel and FRP with the design strength, Young's Modulus and ultimate strain. |
| Loads and reaction forces | Reaction forces and design loads are determined by common loads of a Dutch office building [26,27]. Hollow core slab (10 m) and concrete beams (7.2 m). |
| Bending reinforcement | Calculation of the lever-arm and the concrete compression zone. The thickness of FRP in bending is determined by the design bending moment reduced by the moment resistance of the steel reinforcement. The steel reinforcement is still needed, in order to avoid the occurrence of multiple failure mechanisms, such as concrete crushing. |
| Shear reinforcement | For the shear reinforcement, first, the concrete shear capacity is determined for a hollow beam cross-section and is used as a guideline in further calculations. For the strain, no use is made of $\varepsilon_{fu}$, but a reduced effective strain [$\varepsilon_{f,eff}$] is applied, which includes the effects during cracking and the possible outcome of a failure mechanism. |
| Failure mechanisms | Ultimate Limit State (ULS) failure mechanisms are: steel yielding followed by concrete crushing, steel yielding followed by FRP rupture and concrete crushing. For Serviceability Limit State (SLS) the stress limitation is checked for each material. |
| Deflection | First, the second moment of inertia, with the application of FRP, is determined. Then, the deflection is checked according to EC2 [28,29] recommendations. |

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
