# Peer review of "Environmental Optimization of Precast Concrete Beams Using Fibre Reinforced Polymers"

_sustainability, doi:10.3390/su11072174_

Round 1
Reviewer 1 Report
The paper addresses an important and timely topic– the environmental minimization. The obtained results are interesting. However, the authors should address some issues before publication consideration can be made.
1. The information regarding shadow costs must be completed. Authors should provide a table with this information.
2. Paper talks about life cycle assessment. However, I cannot find the life cycle assessment during use and end of life stages.
3. Besides, the optimization algorithm should be explained. Authors should complete the information regarding optimization procedure.
Reviewer 2 Report
The study is very interesting, and give a good overview of how various types of fibers influence design of single span concrete beam.
But one type of FRP is missing so I would recommend adding in this study basalt fibers (BFRP) see article :
Marianne Kjendseth Wiik, Eythor Rafn Thorhallsson, Kamal Azrague: A Mechanical and Environmental Assessment and Comparison of Basalt Fibre Reinforced Polymer (BFRP) Rebar and Steel Rebar in Concrete Beams. Energy Procedia 03/2017; 111(C):31-40., DOI:10.1016/j.egypro.2017.03.005
There are a few things I would like to clear better:
The colors in figure 1 are to like each other, so the picture is not quite clear as it is now.
In figure 3 is shown stresses in concrete and in FRP, but for what case are these values?
Also as the FRP tubes are used, there are also stresses in that material. How are stresses from the tube included in the overall design?
Also in figure 3, there are no size measurements of the beams.
In table 2, the should be also a column with the regular concrete beam as a reference.
Round 2
Reviewer 1 Report
Because the authors have addressed all my comments, the paper worth publishing in this journal in my opinion.